# A 320,000-year-old blue ice identified at the surface of the Elephant Moraine region, East Antarctica

Giyoon Lee[1], Jinho Ahn[1*], Hyeontae Ju[2], Ikumi Oyabu[3,4], Florian Ritterbusch[5], Songyi Kim[6], Jangil Moon[6], Joohan Lee[2], Yeongcheol Han[6], Soon Do Hur[6], Kenji Kawamura[3,4,7], Zheng-Tian Lu[5,8], Wei Jiang[5,8] and Guo-Min Yang[5,8]

[1]School of Earth and Environmental Sciences, Seoul National University, Seoul, 08826, South Korea

[2]Center of Technology Development, Korea Polar Research Institute, Incheon, 21990, South Korea

[3]National Institute of Polar Research, Research Organization of Information and Systems, Tachikawa, 190-8518, Japan

[4]Polar Science Program, Graduate Institute for Advanced Studies (SOKENDAI), Tachikawa, 190-8518, Japan

[5]Hefei National Research Center for Physical Sciences at the Microscale and School of Physical Sciences, University of Science and Technology of China, Hefei, 230026, China

[6]Division of Glacier & Earth Sciences, Korea Polar Research Institute, Incheon, 21990, South Korea

[7]Japan Agency for Marine-Earth Science and Technology (JAMSTEC), Yokosuka, 237-0061, Japan

[8]Hefei National Laboratory, University of Science and Technology of China, Hefei, 230088, China

*Correspondence to*: Jinho Ahn (jinhoahn@snu.ac.kr)

**Abstract.** For addressing important paleoclimatic questions, such as the cause of the Mid-Pleistocene Transition (MPT), the search for one-million-year-old ice is of great interest. Antarctic blue-ice areas (BIAs), where ancient ice outcrops on the surface of ice sheet, offer promising sites for identifying ice spanning the MPT period. To date, only two sites, the Allan Hills BIA and the Mullins Glacier in East Antarctica, have been identified as areas that contain ancient ice older than one million years. We investigated icefields in the Elephant Moraine and Reckling Moraine regions of East Antarctica to contribute to the search for ancient ice spanning the MPT. Ice-penetrating radar surveys revealed that ice thickness ranged from 200 m to 800 m across the icefields. The $^{81}$Kr dating of the surface ice (<10 m) showed ages of 83–119 kyr BP (Before Present) and 93–124 kyr BP for blue ice in the Meteorite City Icefield and 320–385 kyr BP in the Elephant Moraine Main Icefield. We also analyzed several gas compositions ($\delta^{15}$N-N$_2$, $\delta^{18}$O-O$_2$, $\delta$O$_2$/N$_2$, $\delta$Ar/N$_2$, CO$_2$, CH$_4$, and N$_2$O) and revealed that gas records at very shallow depths are altered. A comparison of stable water isotopes ($\delta^{18}$O$_{ice}$ and $\delta^{2}$H$_{ice}$) indicated that the original deposition site of the Elephant Moraine Main Icefield experienced colder condition than those of the nearby icefields. Given these findings, ice spanning the MPT period could be retrieved from the Elephant Moraine Main Icefield with only a few hundred meters of drilling.

# 1 Introduction

Glacial ice in the polar ice sheets is formed by the compaction of accumulated snow. During this densification process, the air in the firn becomes gradually isolated and trapped as bubbles within the ice, thereby serving as an invaluable archive of ancient atmospheric air (Schwander and Stauffer, 1984). Glacial ice then flows toward the margins of the ice sheet under the influence of gravity. When it encounters topographic obstacles such as nunataks, the ice flow is redirected and thereafter outcrops at the surface of the ice sheet in so-called blue-ice areas (BIAs) (Bintanja, 1999; Sinisalo and Moore, 2010; Gardner

et al., 2018). The total area of BIAs in Antarctica is estimated to be 234,549 $km^2$, accounting for approximately 1.67 % of the Antarctic continent (Hui et al., 2014). Ice layers of the same age are extended to the surface of the BIAs. As a result, virtually unlimited amounts of ancient ice of specific ages can be obtained cost-effectively in BIAs compared to conventional deep-ice-core-drilling projects. In addition, easily accessible old ice in BIAs offers a valuable testbed for developing and applying novel exotic tracers that are currently too risky or impractical to use in conventional deep ice cores.

The 800,000-year-old EPICA Dome C (EDC) ice core, the oldest continuous ice core, has contributed significantly to past atmospheric air composition reconstructions and enhanced our understanding of the Earth's climate system (EPICA community members, 2004; Loulergue et al., 2008; Extier et al., 2018). Nevertheless, to address important questions—such as the cause of the Mid-Pleistocene Transition (MPT), when glacial-interglacial cycles changed from a 40,000-year to a 100,000-year cycle approximately one million years ago—ongoing efforts aim to retrieve ice cores older than one million

45  years (Fischer et al., 2013; Lilien et al. 2021). Shallow ice core drilling in Allan Hills BIA has also been conducted as part of this initiative (Yan et al., 2019; Higgins et al., 2025).

Several BIAs have been dated using various methods such as ice flow modeling, radiometric analysis, and the synchronization of glaciochemical and/or gas records with well-dated ice cores (Moore et al., 2006; Dunbar et al., 2008; Lee et al., 2022; Hu et al., 2024). The estimated age of ice in Antarctic BIAs range from thousands to millions of years (Table 1).

The oldest blue ice is found in the Allan Hills BIA, where the surface ice age ranges from 90 kyr BP to 250 kyr BP (Before Present) (Spaulding et al., 2013), and ice at depths of 200 m dates back to approximately 6 Myr BP (Higgins et al., 2025). Although the ice stratigraphy in the Allan Hills BIA is substantially disordered, it has provided snapshots of past atmospheric oxygen and greenhouse gas (GHG) variations from the pre-MPT period (Yan et al., 2019; Yan et al., 2021). Very old ice has also been identified in rock glaciers in Antarctica (Table 1). For example, ice at depths of 3–32 m from the Mullins Glacier

has been dated to 1.6 Myr BP (Yau et al., 2015) and ice found in Beacon Valley, which is downstream of Mullins Glacier, up to 8.1 Myr BP (Marchant et al., 2002) (Table 1). The estimated gas age of ice at Mullins Glacier is considered a lower bound because the analyzed air likely represents a mixture of ancient and recent atmosphere (Yau et al., 2015). The age constraint for ice in Beacon Valley, based on $^{40}Ar/^{39}Ar$ tephra dating, has been questioned due to the possibility for reworking and re-transportation of the tephra. Based on the discovery of pre-MPT ice in the Allan Hills BIA and Mullins Glacier, Antarctica

may provide additional promising sites for recovering such an ice core by shallow drilling. To identify potential sites, chronological studies of the surface ice must first be conducted.

In this study, we investigated icefields in the Elephant Moraine (EM) (76.32° S, 157.20° E) and Reckling Moraine (RM) (76.24° S, 158.39° E) regions, focusing primarily on constraining the age of blue ice in the EM region (Fig. 1). We first began with assessing the bedrock topography and ice thickness using ice-penetrating radar (IPR) surveys. Next, we took the measurements of trapped air (e.g. $^{81}$Kr, $^{85}$Kr, $\delta^{15}$N-$N_2$, $\delta^{18}$O-$O_2$, $\delta O_2/N_2$, $\delta Ar/N_2$, $CO_2$, $CH_4$, and $N_2O$) in EM blue ice and stable water isotopes ($\delta^{18}O_{ice}$ and $\delta^2H_{ice}$) of the RM and EM blue ice. Finally, considering potential alterations in the measured gas components, we determined the age of the EM blue ice through $^{81}$Kr dating and chemical analyses of the trapped air.

**Table 1.** Age constraints of Antarctic blue-ice areas (BIAs) and rock glaciers[*].

| Blue-ice areas | Age (kyr BP) | Location | References |
| --- | --- | --- | --- |
| Meteorite City Icefield | 101, 108 | 76.25° S, 156.56° E | This study |
| Elephant Moraine Main Icefield | 320 | 76.32° S, 157.20° E | This study |
| Allan Hills | 90–250, 2700, 6000 | 76.73° S, 159.36° E | Spaulding et al. (2013), Yan et al. (2019), Higgins et al. (2025) |
| Frontier Mountain | <50 | 72.98° S, 160.33° E | Folco et al. (2006) |
| Grove Mountains | 143 | 72.99° S, 75.22° E | Hu et al. (2024) |
| Larsen Glacier | 6–25 | 74.93° S, 161.60° E | Lee et al. (2022) |
| Mt. Moulton | 105–136, 496 | 76.67° S, 134.70° W | Dunbar et al. (2008), Korotkikh et al. (2011) |
| Mullins Glacier[*] | 1600 | 77.88° S, 160.54° E | Yau et al. (2015) |
| Beacon Valley[*] | 8100 (?) | 77.85° S, 160.59° E | Marchant et al. (2002) |
| Nansen | <130 | 72.75° S, 24.50° E | Zekollari et al. (2019) |
| Patriot Hills | 1–80, 130–134 | 80.30° S, 81.35° W | Turney et al. (2020) |
| Scharffenbergbotnen | <11 | 74.56° S, 11.05° W | Sinisalo et al. (2007) |
| South Yamato | 55–61 | 72.08° S, 35.18° E | Moore et al. (2006) |
| Taylor Glacier | 9–133 | 77.75° S, 161.80° E | Buizert et al. (2014) |

## 2 Study area and methods

### 2.1 Study area

During the 2016/17 austral summer, shallow ice cores (5–10 m in length and 10 cm in diameter) were collected from the icefields within the EM region (Fig. 1) (Jang et al., 2017). In this study, three shallow cores (EM-B, EM-C, and EM-K) were used for gas analyses ($^{81}$Kr, $^{85}$Kr, $\delta^{15}$N-$N_2$, $\delta^{18}$O-$O_2$, $\delta O_2/N_2$, $\delta Ar/N_2$, $CO_2$, $CH_4$, and $N_2O$). Refer to Sect. 2.3 for Kr measurements, Sect. 2.4 for isotopic ratio measurements of major gas components, and Sect. 2.5 for greenhouse gas concentration measurements, respectively. Additionally, during the 2018/19 austral summer, 70 surface ice samples (5–10 cm depth) were collected along a 700 m transect at 10 m intervals from the icefield in the RM region and were analyzed for stable

water isotopes (Fig. 1) (Sect. 2.6). Ice samples were stored at Seoul National University (SNU) and Korea Polar Research Institute (KOPRI) at −20 °C until analysis.

The EM and RM regions are renowned as Antarctic meteorite stranding zones (Cassidy et al., 1992). The EM region consists

of several distinct icefields: the Northern Ice Patch, Meteorite City Icefield, Texas Bowl Icefield, and the Elephant Moraine Main Icefield (Fig. 1) (Righter et al., 2021). The tephra layer observed in the Meteorite City Icefield (76.25°S, 156.56°E) indicate that the dip of the ice layers range from 20° to 60° (Jang et al., 2017). The mean annual temperature in the EM region is −30.3 °C (Lee et al., 2022), and the annual ablation rate is estimated at $4.1 \pm 0.1$ cm yr$^{-1}$, with a slightly higher rate of $4.7 \pm 0.2$ cm yr$^{-1}$ at RM (Faure and Buchanan, 1991). Ice flows from southwest to northeast at a speed of approximately 1–5 m yr$^{-1}$

(Rignot et al., 2011; Mouginot et al., 2012) (Fig. 1).

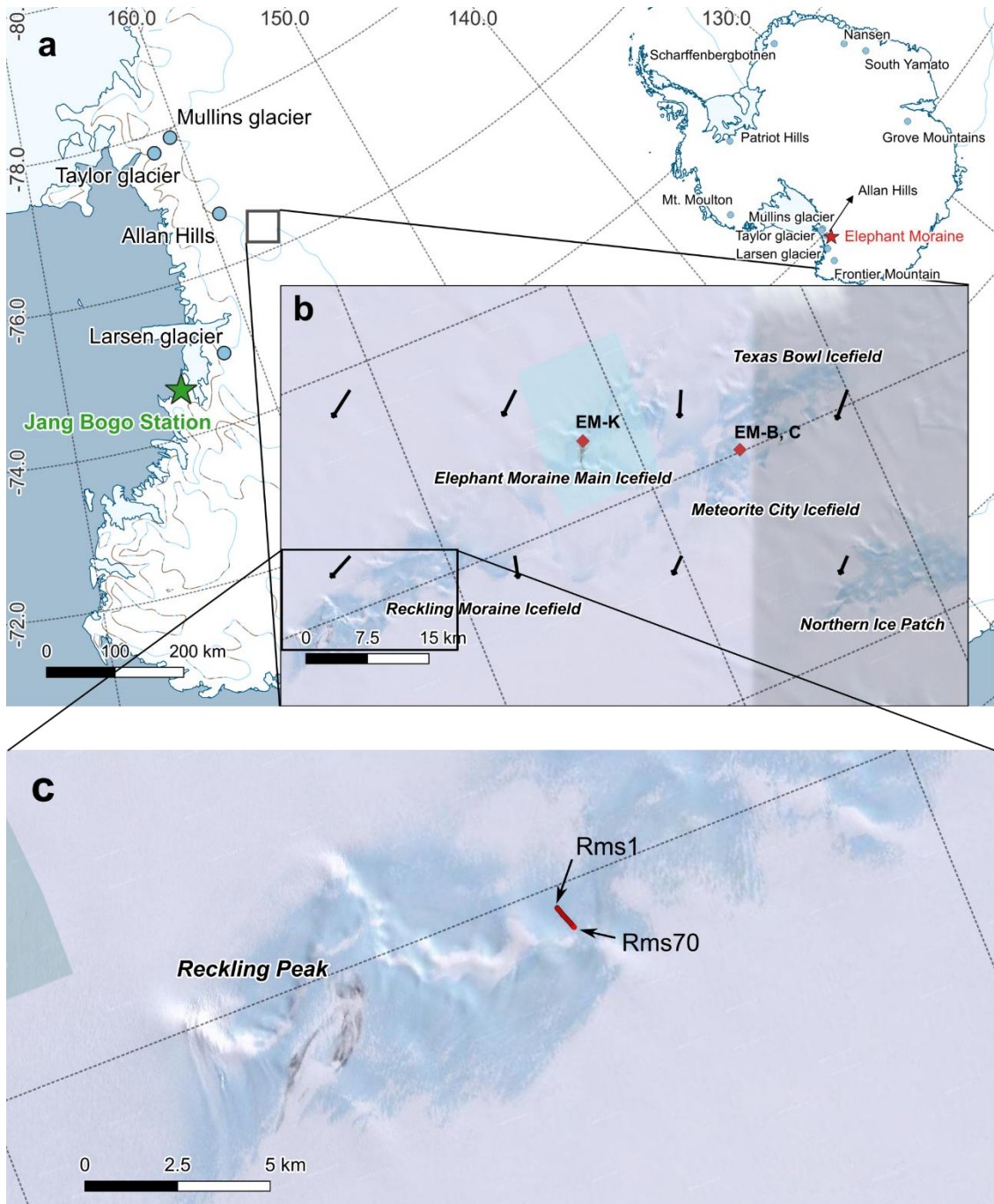

**Figure 1.** Map of Elephant and Reckling Moraine regions. **(a)** The area of Victoria Land, East Antarctica and sites of BIAs. **(b)** Magnified area, including sampling locations of ice core (red diamond). Arrows show the ice flow direction. **(c)** Magnified area, including sampling locations of surface ice (red circles). The map was made using the QGIS Quantarctica package with a satellite image from © Google Earth
(Rignot et al., 2011; Mouginot et al., 2012).

## 2.2 Ice-Penetrating Radar (IPR) survey

During the 2018/19 austral summer, an ice-penetrating radar (IPR) survey was conducted across the EM and part of the RM region to estimate the bedrock elevation and ice thickness. The survey was performed using airborne IPR with 5 km grid spacing, covering a total distance of 384 km. The Helicopter-borne Radar (HERA) system, developed by the University of Texas Institute for Geophysics, was mounted inside a helicopter with two radar boom antennas. The helicopter maintained a constant operating speed of approximately 36 m s$^{-1}$ throughout the survey. Data were recorded at 3200 samples per trace, with a 20 ns sampling interval and a total recording time of 64 μs. The x-axis resolution was approximately 9 m per trace, and the y-axis resolution was approximately 1.69 m per sample.

The surface elevation ($z_s$) of the ice was calculated by subtracting the flight height above the surface ($h_l$), measured using a laser altimeter, from the flight altitude ($z_{hf}$), recorded by the Global Navigation Satellite System (GNSS) onboard the helicopter (Eq. (1)). The ice thickness ($h_{ice}$) was determined by identifying the air-ice and ice-bedrock interfaces in the radar profiles and multiplying the two-way travel time by the radar wave velocity in ice ($v_{ice}$ = 0.169 m ns$^{-1}$, Reynolds, 2011). The bedrock elevation ($z_{bed}$) was estimated by subtracting the ice thickness from the surface elevation (Eq. (2)). The bedrock elevation and ice thickness between survey lines were interpolated using the Kriging method.

$$z_s = z_{hf} - h_l \tag{1}$$

$$z_{bed} = z_s - h_{ice} \tag{2}$$

## 2.3 $^{81}$Kr dating

For $^{81}$Kr dating, 6–10 kg of ice was used for each measurement (Table 2). Because of ice core availability, we mixed different depth ranges for EM-B and EM-K (Table 2). Trapped air was extracted using an ice melter described by Tian et al. (2019). The ice samples were placed in a stainless-steel tank, which was pre-evacuated using a dry scroll pump equipped with a water trap. The ice was then melted by immersing the tank in hot water to release the trapped gas. The extracted gas was collected in a stainless-steel container and transported to the University of Science and Technology of China (USTC) for Kr purification and $^{81}$Kr analysis. $^{81}$Kr analysis was performed using the Atom Trap Trace Analysis (ATTA) method, and $^{85}$Kr was also measured to quantify the potential contamination from modern air, following Tian et al. (2019) and Jiang et al. (2012).

**Table 2.** Results of Kr analysis of shallow ice cores from Elephant Moraine region. TAC: total air content, STP: standard temperature and pressure, dpm cm$^{-3}$: decay per minute per cubic centimeter of Kr, pMKr: percent modern krypton.

| Sample | Extracted air (mL) | Mass (kg) | TAC (cm$^3$ STP g$^{-1}$) | Depth (cm) | $^{85}$Kr (dpm cm$^{-3}$) | $^{81}$Kr (pMKr) | $^{81}$Kr age (kyr BP) | Systematic error (kyr) |
|---|---|---|---|---|---|---|---|---|

| | | | | | | | | |
|---|---|---|---|---|---|---|---|---|
| EM-B | 432 | 9.3 | 0.043 | 515–762.5, 795.5–967.5 | <1.0 | 74.1 ± 3.9 | $101^{+17}_{-17}$ | ± 4.9 |
| EM-C | 517 | 10.1 | 0.048 | 271.5–680.5 | <0.8 | 72.4 ± 3.3 | $108^{+15}_{-14}$ | ± 5.2 |
| EM-K | 425 | 6.4 | 0.061 | 151–283.5, 329.5–384.5, 464.5–568 | 5.2 ± 0.4 | 39.0 ± 2.6 | $351^{+29}_{-26}$ | ± 16.9 |
| Air in Seoul (Nov. 2019) | - | - | - | | 78.9 ± 1.9 | - | - | - |

## 2.4 $\delta^{15}$N-N$_2$, $\delta^{18}$O-O$_2$, $\delta$O$_2$/N$_2$, and $\delta$Ar/N$_2$

Based on ice core availability, six ice samples were cut from the EM ice cores and sent to the National Institute of Polar Research (NIPR) in Japan on December 2019 for the simultaneous measurement of O$_2$, N$_2$ isotopes, and O$_2$, N$_2$, Ar molecular ratios using a dual-inlet mass spectrometer (Thermo Fisher Delta V) following Oyabu et al. (2020). The reproducibility for $\delta^{15}$N-N$_2$, $\delta^{18}$O-O$_2$, $\delta$O$_2$/N$_2$, and $\delta$Ar/N$_2$ are 0.006 ‰, 0.011 ‰, 0.09 ‰, and 0.12 ‰, respectively (Oyabu et al., 2020). Until analysis on December 2022, the ice samples were stored at NIPR at around −30 °C. The outermost surface and any large cracks were carefully trimmed by approximately 3–5 mm, and blurry ice surfaces were shaved off using a ceramic knife. The final mass of the sample was 70–130 g. The ice was then loaded into a stainless-steel vessel, and the trapped gas was released by immersing it in hot water. The released gas was cryopumped, passed through a water trap, and finally collected in a stainless-steel tube for analysis.

The isotope ratios of gases in ice cores can be affected by fractionation owing to gravitational settling and thermal diffusion in the firn column (Craig et al., 1988; Severinghaus et al., 1998; Goujon et al., 2003). To correct for gravitational fractionation, we applied Eq. (3).

$$\delta_{\mathrm{grav}} = \delta_{\mathrm{measured}} - (\Delta m \times \delta^{15}\mathrm{N}) \tag{3}$$

$\Delta m$ represents the mass difference between heavy and light isotopes; 2 for $\delta^{18}$O-O$_2$ ($^{18}$O/$^{16}$O), 4 for $\delta$O$_2$/N$_2$ ($^{32}$O$_2$/$^{28}$N$_2$), and 12 for $\delta$Ar/N$_2$ ($^{40}$Ar/$^{28}$N$_2$). We assumed that thermal fractionation correction was not necessary because the relatively gradual climate change in Antarctica is unlikely to induce a significant temperature gradient within the firn column (Severinghaus et al., 1998; Goujon et al., 2003).

Gas loss can occur due to storage temperatures above −50 °C and/or the presence of numerous fractures in the ice, leading to depletion in $\delta$O$_2$/N$_2$ values and enrichment in $\delta^{18}$O-O$_2$ values in bubbly ice (Bender et al., 1995; Ikeda-Fukazawa et al., 2005; Severinghaus et al., 2009). However, due to insufficient measurements for gas loss correction, we could not apply gas loss correction in this study (Landais et al., 2003; Capron et al., 2010; Baggenstos et al., 2017). The final isotope ratios are reported relative to those in the modern atmosphere.

## 2.5 Greenhouse gas concentrations ($CO_2$, $CH_4$, and $N_2O$)

The $CO_2$ concentrations in the EM blue ice (EM-B, EM-C, and EM-K) were measured at SNU following Shin (2014) and Lee et al. (2022). To eliminate potential contamination from ambient air, the outermost surface and large cracks of the ice were carefully trimmed to approximately 1–2 mm using a band saw. Ice (15–20 g) was then placed in a double-walled vacuum chamber maintained at −35 °C during sample preparation. The trapped ancient air within the ice was released using a needle crusher, cryopumped through a −85 °C water trap, and finally condensed in stainless-steel tubes at 12 K (−261 °C). The tubes were then warmed up in hot water and attached to a flame ionization detector gas chromatograph (FID-GC) to measure the $CO_2$ concentration. For this, we used a Ni-catalyst to convert $CO_2$ into $CH_4$ before reaching the detector (Ahn et al., 2009; Shin et al., 2022). The uncertainty of the $CO_2$ concentration measurement is defined as the standard deviation of the $CO_2$ measurement results from the control group (average of intra-day standard deviation of the control group: $0.6 \pm 0.6$ ppm).

The $CH_4$ concentrations in the EM blue ice (EM-B, EM-C, and EM-K) were also measured at SNU using the methods of Yang (2019). Following the same pretreatment process as that used for the $CO_2$ measurements, 45–56 g of ice was placed in a custom-made glass flask. The flask was evacuated for an hour before immersion in hot water to release the trapped air. To prevent $CH_4$ dissolution, the melted water was refrozen by immersing it in a −80 °C ethanol bath. Finally, the extracted air was analyzed using FID-GC to measure the $CH_4$ concentrations. The uncertainty of the $CH_4$ concentration measurement is defined as the standard deviation of the $CH_4$ measurement results from the control group (average of intra-day standard deviation of the control group: $3.3 \pm 1.4$ ppb).

The concentrations of $CO_2$, $CH_4$, and $N_2O$ in several ice core samples were also measured along with gas isotopes ($\delta^{15}N$-$N_2$, $\delta^{18}O$-$O_2$, $\delta O_2/N_2$, and $\delta Ar/N_2$) at the NIPR using wet extraction method following Oyabu et al. (2020). After gas extraction, the gas was split into two aliquots: one for isotope analysis and the other for GHG concentration measurement. $CO_2$ and $CH_4$ concentrations were measured using FID-GC, while $N_2O$ concentration was determined with an electron capture detector (ECD) GC.

## 2.6 Stable water isotopes

Stable water isotopes ($\delta^{18}O_{ice}$ and $\delta^2H_{ice}$) of the surface RM blue ice (approximately 5–10 cm depth) and EM-K core were measured at the Korea Polar Research Institute (KOPRI) using cavity ring-down spectroscopy (CRDS, Picarro L2130-i) (Fig. 1) following Lee et al. (2022). Ice samples were melted at room temperature and injected into 2 mL vials using disposable syringes equipped with 0.45 µm filters. Measurement precision was achieved by repeatedly measuring the working standard, resulting in a $1\sigma$ (standard deviation) of 0.07 ‰ for $\delta^{18}O_{ice}$ and 0.60 ‰ for $\delta^2H_{ice}$. Water isotope values were calibrated using the international standards of VSMOW2, SLAP2 (Standard Light Antarctic Precipitation 2), and GISP (Greenland Ice Sheet Precipitation).

 **3 Results**

**3.1 Bedrock elevation and ice thickness**

The IPR survey revealed a detailed profile of bedrock topography and ice thickness in the EM region and partially in the RM region (Fig. 2). The maximum bedrock elevation reached approximately 1,600 m above mean sea level (AMSL) in both regions, forming a steep and narrow valley between the icefields (Fig. 2a). The ice thickness ranged from 200 m to 800 m within the icefields and from 1,000 to 1,900 m in the valleys (Fig. 2b). Notably, a local bedrock high, approximately 1,000 m AMSL was identified between the Elephant Moraine Main Icefield and the RM regions. This topographic feature may act as a barrier for ice flowing from the EM to RM region.

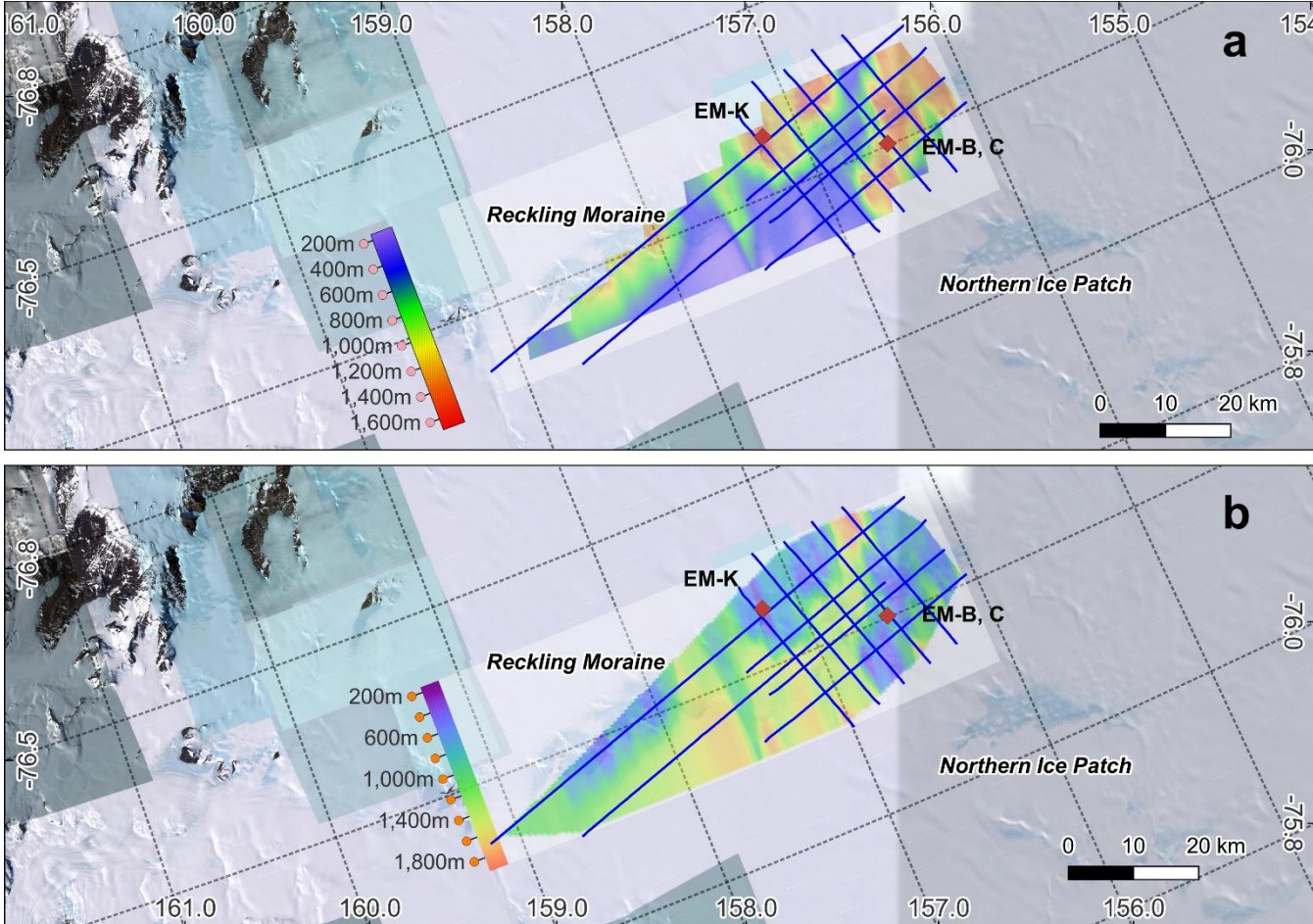

**Figure 2. (a)** Bedrock elevation (AMSL) and **(b)** ice thickness of Elephant and Reckling Moraine regions. The blue lines are the survey line of IPR. Red diamonds are the locations of the shallow ice cores drilled. The map was made using a satellite image from © Google Earth in QGIS.

## 3.2 $^{81}$Kr dating

The measured $^{85}$Kr activity, a proxy for modern air contamination in the EM-B and EM-C cores was below the detection limit (90 % confidence level), indicating that correction for modern air contamination was not necessary. The resulting $^{81}$Kr ages were $101^{+17}_{-17}$ kyr BP and $108^{+15}_{-14}$ kyr BP for the analyzed samples of EM-B and EM-C cores, respectively (Table 2). Different from the EM-B and EM-C samples, the air extracted from the EM-K core exhibited $^{85}$Kr activity of $5.2 \pm 0.4$ (dpm cm$^{-3}$), indicating slight contamination by modern air (Table 2). After modern air correction, assuming that modern air from Seoul was the contamination source, the $^{81}$Kr age of the analyzed samples of EM-K core was $351^{+29}_{-26}$ kyr BP (Table 2). It is important to note that $^{81}$Kr ages were given with statistic uncertainties ($1\sigma$ confidence level) because of atom counting. Additionally, a systematic error arises from the half-life of $^{81}$Kr ($229 \pm 11$ kyr) and variations in the past atmospheric $^{81}$Kr abundance (Zappala et al., 2020). Considering these uncertainties, the age ranges of analyzed samples of EM-B and EM-C cores were 83–119 kyr BP and 93–124 kyr BP, respectively. The age range of analyzed samples of the EM-K core was 320–385 kyr BP. Since the $^{81}$Kr dating was conducted in 2020, its reference point is 2020. This results in a 0.07 kyr difference from the kyr BP notation, which uses 1950 as the reference year. However, this difference was considered negligible in this study.

## 3.3 $\delta^{15}$N-N$_2$, $\delta^{18}$O$_{atm}$, $\delta$O$_2$/N$_2$, and $\delta$Ar/N$_2$

Ice cores from the Meteorite City Icefield (EM-B and EM-C) and the Elephant Moraine Main Icefield (EM-K) showed distinct values for $\delta^{15}$N-N$_2$, $\delta^{18}$O$_{atm}$, $\delta$O$_2$/N$_2$, and $\delta$Ar/N$_2$ (Table 3). The $\delta^{15}$N-N$_2$ values in the EM-B and EM-C cores were lower compared to those in the EM-K core, which may be attributed to a thinner diffusive zone in the firn at the original deposition site of the Meteorite City Icefield compared to that of the Elephant Moraine Main Icefield. The $\delta^{18}$O$_{atm}$ values in the EM-K core were also lower than those in the EM-B and EM-C cores and included a negative value of −0.105 ‰ (Table 3), which indicates a relatively warm period.

The $\delta$O$_2$/N$_2$ and $\delta$Ar/N$_2$ values corrected for gravitational fractionation in the EM-B and EM-C cores exhibited significantly high positive values, similar to blue ice samples from Allan Hills in the upper 15 m of ice cores (Spaulding et al., 2013). However, those in the EM-K core were negative, and slightly less positive (Table 3).

**Table 3.** GHG concentrations and gas isotope ratios of shallow ice from the Elephant Moraine region measured from NIPR using wet extraction method. Only isotopic values were corrected for gravitational fractionation. NA, not available.

| Sample | Depth (cm) | TAC (cm$^3$ STP g$^{-1}$) | CH$_4$ (ppb) | CO$_2$ (ppm) | N$_2$O (ppb) | $\delta^{15}$N-N$_2$ (‰) | $\delta^{18}$O$_{atm}$ (‰) | $\delta$O$_2$/N$_2$ (‰) | $\delta$Ar/N$_2$ (‰) |
|---|---|---|---|---|---|---|---|---|---|
| EM-B | 659.0–669.0 | 0.053 | 518.9 | 272.8 | 257.7 | 0.248 | 0.554 | 10.312 | 10.434 |
| EM-B | 827.5–837.5 | NA | NA | NA | NA | 0.222 | 0.495 | 32.326 | 24.734 |
| EM-C | 416.5–426.5 | 0.041 | 521.3 | 292.9 | 286.9 | 0.219 | 0.399 | 49.857 | 34.521 |
| EM-C | 558.0–568.0 | 0.055 | 518.7 | 277.0 | 272.3 | 0.237 | 0.503 | 15.978 | 11.626 |

| EM-K | 247.5–257.5 | 0.063 | 485.3 | 312.0 | 294.0 | 0.326 | 0.286 | -1.082 | 0.229 |
| EM-K | 489.5–494.5 | 0.080 | 676.9 | 309.4 | 297.5 | 0.378 | -0.105 | 2.717 | -1.469 |

### 3.4 Greenhouse gases ($CO_2$, $CH_4$, and $N_2O$)

The measured $CO_2$ concentrations in the EM-B and EM-C cores increased toward the surface, exceeding 300 ppm, which is beyond the natural concentration range during the past 800 kyr (180–300 ppm) (Fig. 3, Table S1) (Bereiter et al., 2015). Additionally, $CO_2$ concentrations higher than 300 ppm were identified at depths of 8.8 m in EM-B and 2.8 m in EM-C, respectively (Fig. 3, Table S1). In the EM-K core, a notably high $CO_2$ concentration of 628 ppm at a depth of approximately 0.6 m, which is even greater than modern atmospheric $CO_2$ concentrations, and approximately 350 ppm at 4.7 m depth were identified (Fig. 3, Table S1).

The $CH_4$ concentrations in EM-B and EM-C cores increased toward the surface, exceeding 800 ppb, which is beyond the natural concentration range during the past 800 kyr (340–800 ppb) (Fig. 3, Table S2) (Loulergue et al., 2008). However, at depths greater than 1 m, the measured $CH_4$ concentrations were in alignment with the natural concentration range during the past 800 kyr (Fig. 3, Table S2). In contrast, the $CH_4$ concentrations in the EM-K core showed a decreasing trend toward the surface, reaching a very depleted concentration of 207 ppb, which is lower than the natural concentration range during the past 800 kyr. At a depth of 5 m, the EM-K core exhibited a $CH_4$ concentration of approximately 950 ppb, which is greater than the natural concentration range during the past 800 kyr (Fig. 3, Table S2).

Several measured $N_2O$ concentrations (Table 3) were in line with warm interglacial values, but they might have been affected by in-situ production from dust (Schilt et al., 2014).

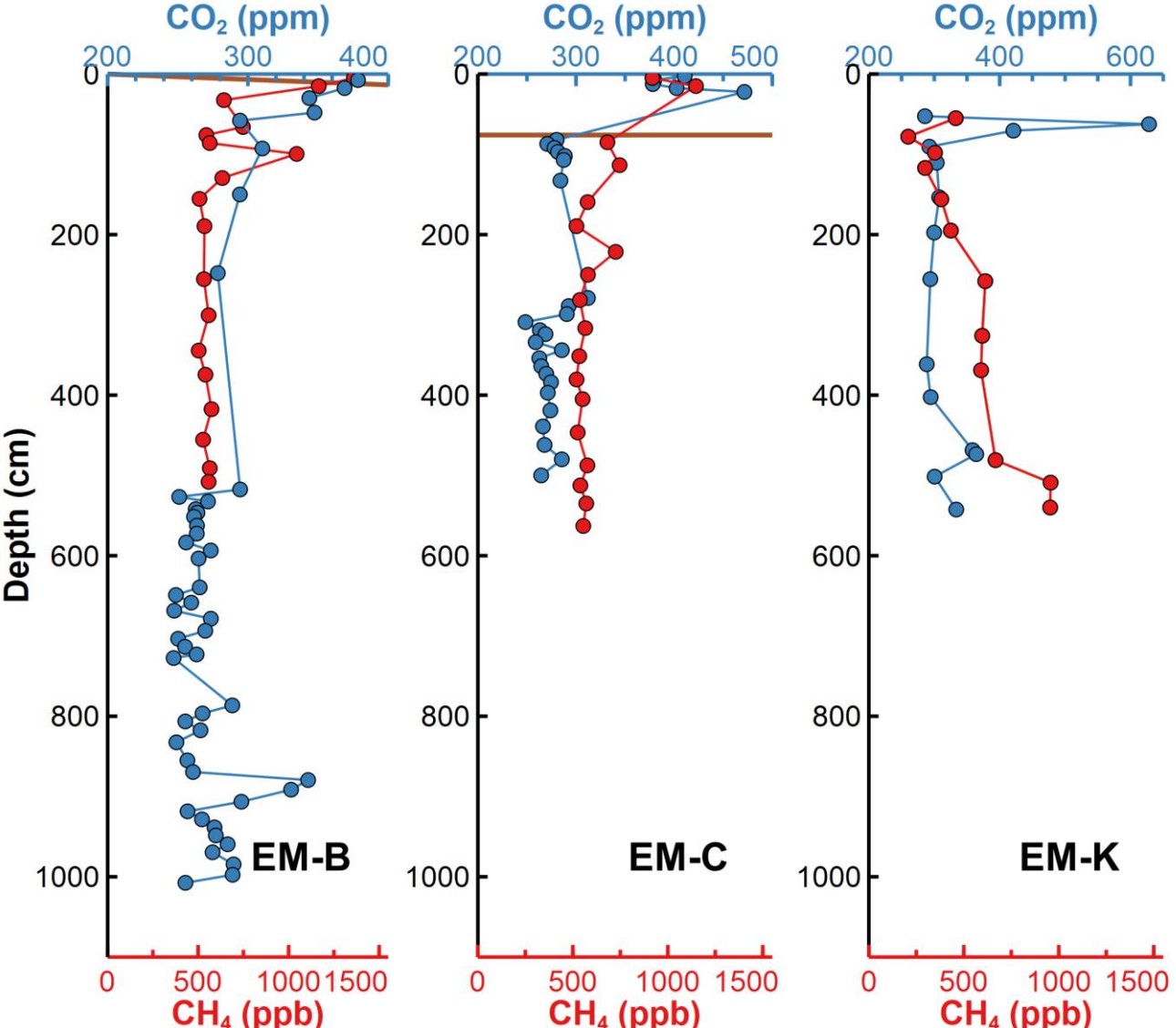

**Figure 3.** Vertical profiles of $CO_2$ and $CH_4$ concentrations in Elephant Moraine blue ice. The $CO_2$ concentration was measured by a dry-extraction system in SNU. The $CH_4$ concentration was measured by a wet-extraction system in SNU. Brown lines indicate the dust band identified in the EM-B and EM-C ice cores. The dust band in EM-B has a dip of 59°, whereas the dust band in EM-C is parallel to the surface (Jang et al., 2017).

### 3.5 Stable water isotopes

The stable water isotope measurements ($\delta^{18}O_{ice}$ and $\delta^2H_{ice}$) of RM blue ice showed no discernible increasing or decreasing trend along the transect (Fig. S1). The average $\delta^{18}O_{ice}$ and $\delta^2H_{ice}$ values of RM blue ice were −38.6 ± 1.4 ‰ (1σ) and −311.0

± 11.8 ‰ (1σ), respectively (Table S3). The average $\delta^{18}O_{ice}$ and $\delta^2H_{ice}$ values of EM-K blue ice were −45.6 ± 0.3 ‰ (1σ) and −362.2 ± 2.6 ‰ (1σ), respectively (Table S4).

Compared to $\delta^{18}O_{ice}$ measurements from ice sampled around Reckling Peak, which ranged from −51.2 ‰ to −41.2 ‰ (Faure et al., 1992), our measured RM blue ice $\delta^{18}O_{ice}$ values were relatively higher, ranging from −42.0 ‰ to −34.9 ‰. Since the typical glacial-interglacial $\delta^{18}O_{ice}$ difference in East Antarctica is 5–6 ‰, based on conventional deep ice cores (Stenni et al., 2010), the very wide $\delta^{18}O_{ice}$ range observed in RM blue ice (from −51.2 ‰ to −34.9 ‰) may suggest significant differences in provenance of blue ice. Alternatively, since the accumulation site of RM region is likely located on the flank of the East Antarctic ice sheet, it may have experienced large changes in surface elevation and/or temperature. Hence, if the provenance is not significantly different, such a wide range could indicate large change in surface elevation and/or temperature.

The EM-K core showed the most negative $\delta^{18}O_{ice}$ values (Fig. 4), suggesting that surface snow at the original deposition site of the Elephant Moraine Main Icefield experienced colder conditions than those at nearby icefields (Texas Bowl and RM blue ice). In contrast, the RM blue ice had the most enriched water isotope values (Fig. 4), suggesting that its origin experienced warmer conditions than those of the Elephant Moraine Main Icefield and Texas Bowl blue ice.

The deuterium excess (d-excess = $\delta^2H_{ice}$ − 8 × $\delta^{18}O_{ice}$) values exhibited negative values in both the RM and Texas Bowl blue ice samples (Table S3) (Jang et al., 2017). RM blue ice showed an average of −1.8 ± 1.5 ‰ (1σ) (Table S3), while Texas Bowl blue ice exhibited even more negative d-excess values, averaging −3.4 ± 1.4 ‰ (1σ) (Fig. 4) (Jang et al., 2017). The d-excess can be influenced by the source of the water vapor, supersaturation during cloud formation, and post-depositional alterations such as sublimation. Recent finding indicated that sublimation significantly contributed to negative d-excess values observed in surface snow and ice near the Dry Valleys in Antarctica (Hu et al., 2022). Therefore, it is plausible that isotope fractionation driven by sublimation also impacted the original deposition sites of RM and Texas Bowl blue ice samples. The EM-K core on the other hand, showed a positive d-excess value of 2.7 ± 1.3 ‰ (1σ), suggesting relatively less isotope fractionation because of sublimation at the original deposition site of Elephant Moraine Main Icefield than those of RM and Texas Bowl Icefields. The significantly different d-excess values in Elephant Moraine Main Icefield may indicate that the provenance of the blue ice differs from that of the other icefields.

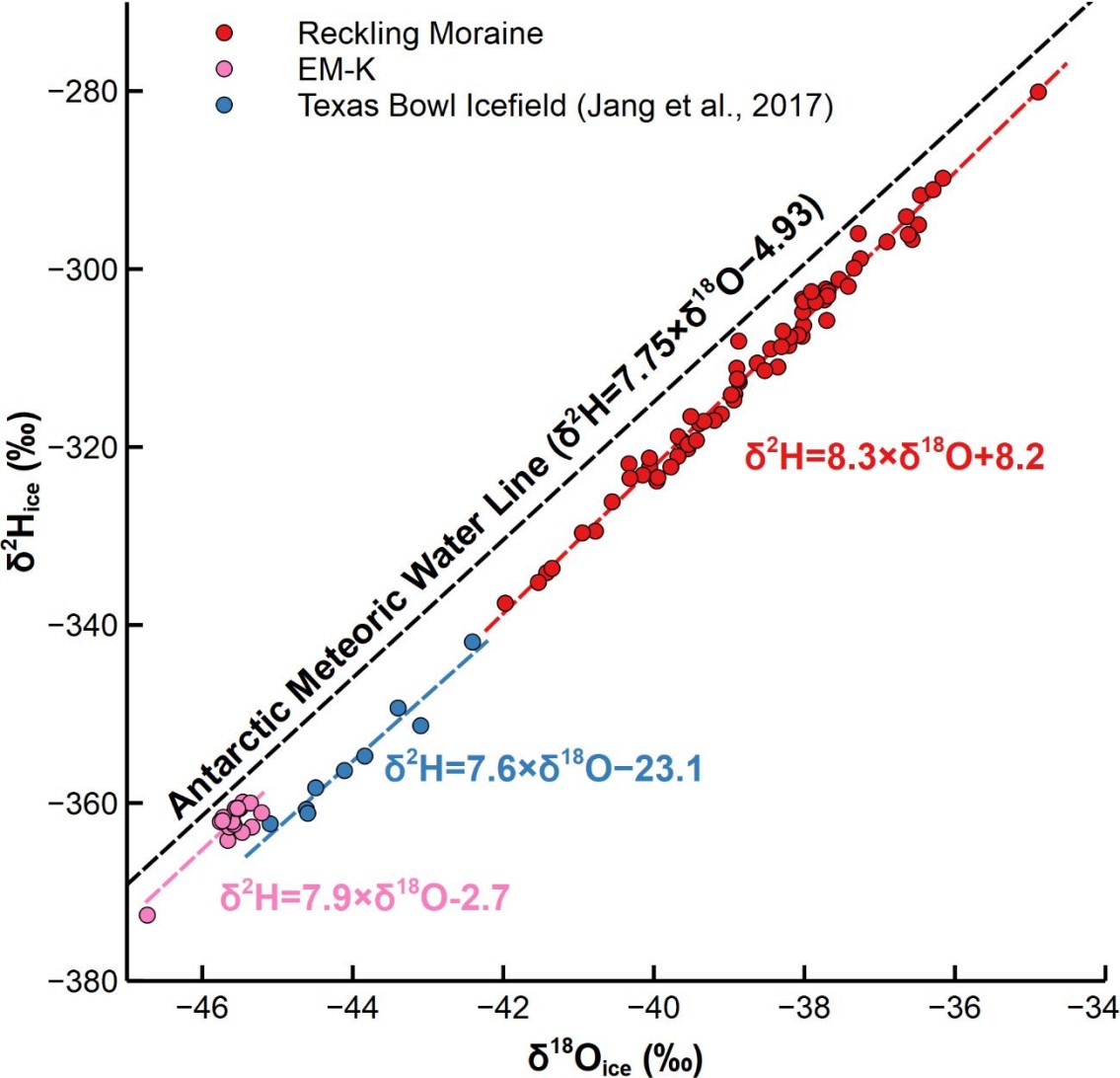

**Figure 4.** Stable water isotope bi-plot comparing the RM blue ice measurements and Elephant Moraine blue ice with the Antarctic Meteoric Water Line (Masson-Delmotte et al., 2008).

## 4 Discussions

### 4.1 Altered gas records in surface EM blue ice

**4.1.1 Relatively low total air content (TAC)**

     The TAC of EM blue ice was derived as a by-product of air sampling for $^{81}$Kr dating at USTC and GHG measurements at NIPR, with values ranging from 0.041 to 0.080 cm$^3$ g$^{-1}$ (Table 2 and 3). These values are substantially lower than those

observed in deep Antarctic ice cores, which typically range from 0.080 to 0.118 $cm^3$ $g^{-1}$ (Raynaud and Whillans, 1982; Martinerie et al., 1994; Delmotte et al., 1999; Raynaud et al., 2007). One reason for the low TAC is partial melting. However, no melt layers were visually observed in the EM ice cores, and the maximum austral summer air temperature of −9.5 °C in the EM region could indicate that partial melting is unlikely (Fig. S2). Furthermore, the original deposition site of the EM blue ice is likely situated further inland, where colder conditions would have prevented surface snow melting. Despite the absence of visible melt layers, line scanning is warranted to detect any potential small-scale melt layers that could result from direct heating by sunlight penetration into blue ice (Cooper et al., 2021). Studies have shown that gas loss during storage occurs when ice core samples were kept at temperatures above –50 °C (Oyabu et al., 2021). After ice core drilling, samples presented in this study were kept at −20 °C or −30 °C for several years until analysis. Gas loss may have also occurred when the ice was exposed at the surface, considering the annual mean temperature of the area is −30.3 °C (Lee et al., 2022). However, gas loss in bubbly ice samples lead to more depleted $\delta O_2/N_2$ and $\delta Ar/N_2$ values, which is contrary to our observations discussed in Sect. 4.1.2.

### 4.1.2 Significantly high positive $\delta O_2/N_2$ and $\delta Ar/N_2$ values

Typical bubbly ice samples from deep ice cores exhibit negative $\delta O_2/N_2$ and $\delta Ar/N_2$ values, ranging from –15 ‰ to 0 ‰ (Landais et al., 2012; Extier et al., 2018; Oyabu et al., 2021). In bubbly ice samples, these values can become even more depleted if gas loss occurs during storage (Oyabu et al., 2021). In this context, the positive $\delta O_2/N_2$ and $\delta Ar/N_2$ values observed in bubbly blue ice samples from EM (Table 3) and Allan Hills BIA are unusual (Spaulding et al., 2013). To explain the positive values observed in the bubbly blue ice samples, Spaulding et al. (2013) suggested two possibilities: (1) the preferential loss of $N_2$ or (2) the addition of $O_2$ and Ar. However, the idea of preferential $N_2$ loss is questionable because $O_2$ and Ar have higher diffusion coefficient than $N_2$ in ice (Ikeda-Fukazawa et al., 2004). The addition of $O_2$ and Ar could have occurred because of surface snow melting at the original deposition site, given that $O_2$ and Ar are more soluble than $N_2$ (Hamme and Emerson, 2004). If then, the measured $\delta^{18}O_{atm}$ value would also be affected. However, it is difficult to determine whether it is depleted or enriched relative to its original value. If the snowmelt refroze after reaching equilibrium with atmospheric air, $^{18}O$ would have preferentially dissolved over $^{16}O$, whereas if it refroze before reaching equilibrium, $^{16}O$ would have dissolved preferentially over $^{18}O$ (Li et al., 2019). However, as discussed in Sect. 4.1.1, surface snow melting at the original deposition site is considered unlikely. Further investigation is needed to understand the positive $\delta O_2/N_2$ and $\delta Ar/N_2$ values observed in blue ice samples.

### 4.1.3 Enriched and depleted GHG concentrations

The enriched GHG concentrations at shallow depths (<1 m) in the EM-B and EM-C cores may be associated with the presence of a visible dust band, observed at depths of 0–13.5 cm in EM-B and 76 cm in EM-C (Jang et al., 2017). A study using blue ice in Pakitsoq, western Greenland also revealed that enriched $CH_4$ values were correlated with visible dust bands, but the mechanism remained unclear (Petrenko et al., 2006). The elevated GHG concentrations around a depth of 5 m in the

EM-K core are attributed to modern air intrusion, as indicated by the $^{85}$Kr activity value (Table 2). If the positive $\delta O_2/N_2$ and $\delta Ar/N_2$ values in EM blue ice result from surface snow melting, the enriched GHG concentrations may also be associated with this process, as $CO_2$ and $CH_4$ are more soluble than the major components of air (Wilhelm et al., 1977). Further investigation is required to better understand the cause of GHG alterations in EM blue ice, particularly the very high $CO_2$ concentration (628 ppm) and unusually low $CH_4$ concentration (207 ppb) in the EM-K core, and the elevated $CO_2$ concentration at depth of approximately 9 m in EM-B.

Altered GHG concentrations in blue ice are also identified in other BIAs in Antarctica and several hypotheses have been suggested to explain the alteration (Turney et al., 2013; Baggenstos et al., 2017; Dyonisius et al., 2023). For example, based on carbon isotopic ratio measurement of $CO_2$, elevated $CO_2$ concentrations have been attributed either to in-situ production from organic compounds or to ice contamination during sampling, transport, and storage (Turney et al., 2013). Another study, which measured carbon isotopic ratio of $CH_4$ has proposed microbial methanotrophic activity as a potential explanation for the low $CH_4$ concentrations observed in blue ice (Dyonisius et al., 2023). Furthermore, elevated GHG concentrations in blue ice could be attributed to microbial activity (Stibal et al., 2012; Baggenstos et al., 2017). However, more rigorous investigation, including analyses of stable isotopes of GHGs, is required to understand its alteration mechanisms in blue ice.

## 4.2 Age constraints of Elephant Moraine blue ice

Based on $^{81}$Kr dating, the shallow ice cores from the Meteorite City Icefield (EM-B and EM-C) correspond to Marine Isotope Stage (MIS) 5, while the shallow ice core from the Elephant Moraine Main Icefield (EM-K) corresponds to MIS 9–11 (Fig. 5). Although the GHG concentrations and $\delta^{18}O_{atm}$ values of the EM blue ice are not pristine, we compared them with unaltered records from Antarctic deep ice cores to further constrain the age of the EM blue ice (Fig. 5). For this comparison, we used the measurement results from depths greater than 3 m, excluding those that fall outside the natural range during the past 800 kyr; we also excluded the elevated GHG concentrations observed at depths of approximately 9 m in EM-B and 5 m in EM-K. Assuming that the original values fall within the average and standard deviation of the measurement results used, EM-B and EM-C cores do not correspond strongly to MIS 5e as the $CH_4$ concentration and $\delta^{18}O_{atm}$ values significantly differ from that period (Fig. 5). Similarly, the ice from the EM-K core is unlikely to correspond to MIS 10–11 and may instead be from early MIS 9, as all three measured gas components ($CO_2$, $CH_4$, and $\delta^{18}O_{atm}$) are consistent with values observed during early MIS 9 (Fig. 5). We consider the measurements from the EM-K core to be more reliable than those from the EM-B and EM-C cores, as TAC values are closer to that of typical deep Antarctic ice cores (Table 3).

The Elephant Moraine Main Icefield contains older surface ice than the Allan Hills BIA, where the surface ice age ranges 90–250 kyr BP (Spaulding et al., 2013). Considering that an ice age of 6 Myr BP has been identified at a depth of 200 m in the Allan Hills BIA (Higgins et al., 2025), the Elephant Moraine Main Icefield presents a strong potential for preserving ancient ice from the MPT period at ice depths of a few hundred meters.

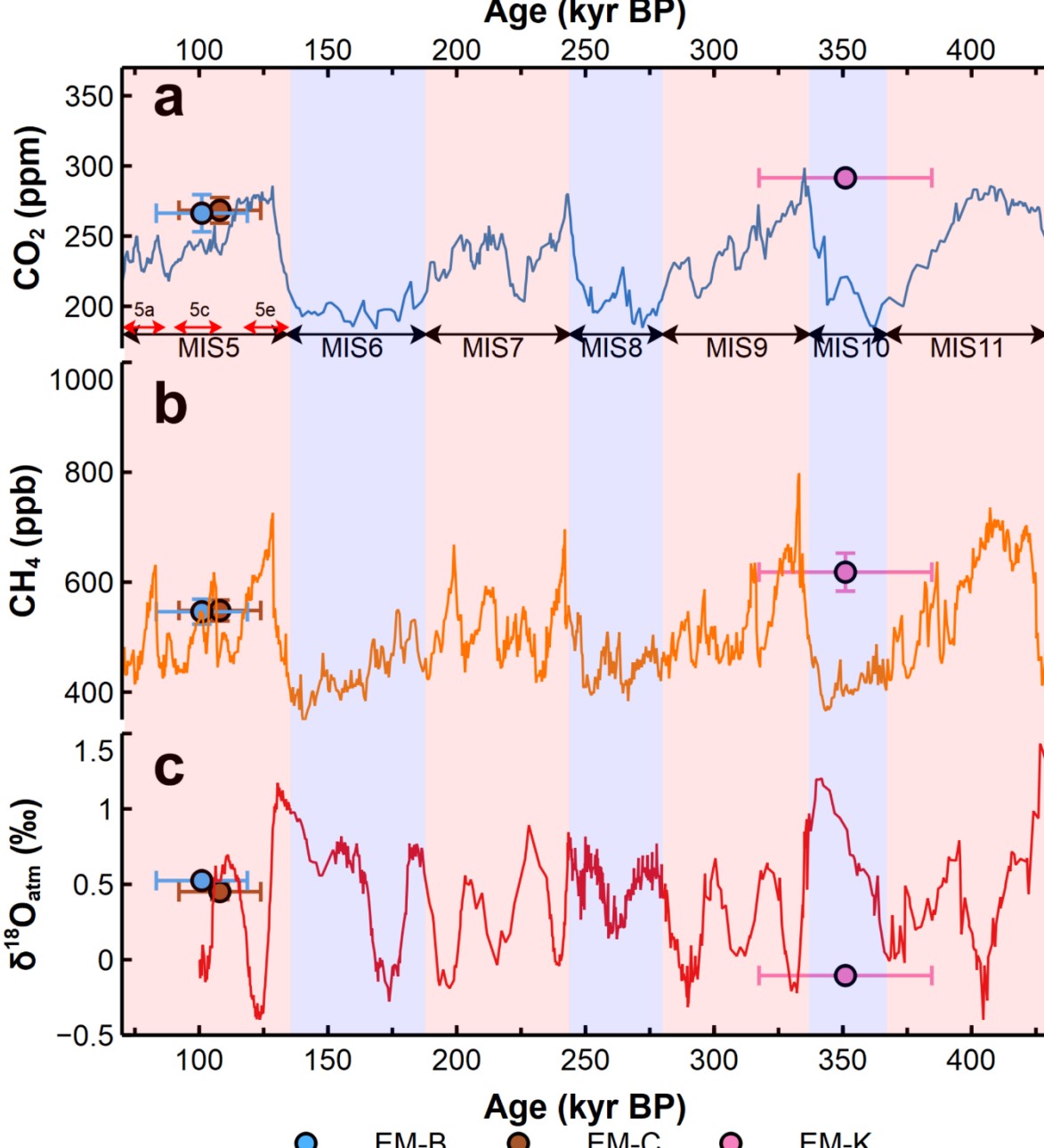

**Figure 5.** Gas composition comparison between EM blue ice and published records from Antarctic ice cores. Horizontal bars for age are the full range of [81]Kr age of analyzed samples of EM-B, EM-C, and EM-K core, considering systematic error together (Table 2). Error bars for $CO_2$, $CH_4$, and $\delta^{18}O_{atm}$ represent $1\sigma$ standard deviation of the measurement result used. **(a)** Blue line is the $CO_2$ composite data from Bereiter et al. (2015). The Marine isotope stage (MIS) numbers are written at the bottom of the panel (Railsback et al., 2015). **(b)** Orange line is the

CH$_4$ concentrations measured from the EDC core (Loulergue et al., 2008). **(c)** Red line is the $\delta^{18}O_{atm}$ measured from the EDC core (Extier et al., 2018).

## 5 Conclusions

In this study, we investigated the blue ice in the Elephant Moraine and Reckling Moraine regions of East Antarctica. The IPR survey revealed that the bedrock elevations reached approximately 1,600 m, while the ice thickness ranged from 200 m to 800 m across the icefields. The $^{81}$Kr dating results indicated ages of 83–119 kyr BP and 93–124 kyr BP for blue ice in the Meteorite City Icefield and 320–385 kyr BP for blue ice in the Elephant Moraine Main Icefield. The comparison of stable water isotopes indicated that the original deposition site of the Elephant Moraine Main Icefield experienced colder condition than those of the Texas Bowl and Reckling Moraine Icefield. Stable water isotope measurements of blue ice in the Reckling

Moraine region showed negative deuterium excess values, indicating that the surface snow at its original deposition site experienced isotope fractionation. The blue ice in the Elephant Moraine region exhibited a very low TAC along with positive $\delta O_2/N_2$ and $\delta Ar/N_2$ values, the causes of which are not yet clearly understood. Additionally, the measured GHG concentrations showed significant alterations, possibly related to high dust content and modern air contamination. Further age constraints, based on comparisons of $CO_2$, $CH_4$, and $\delta^{18}O_{atm}$ measurements with those from Antarctic deep ice cores, suggested that the

surface blue ice in the Elephant Moraine Main Icefield may correspond to early MIS 9. Although pristine gas records are required for more accurate age constraints, we suggest that the Elephant Moraine Main Icefield is a highly promising area for discovering ancient ice spanning the Mid-Pleistocene Transition period.

### Data availability

All data are presented in the main text and in the supplement.

**Author contribution**

GL conceived the idea of this study, measured greenhouse gas concentrations, and wrote the manuscript with contributions from all co-authors. JA conceived the idea of this study and interpreted the data. HJ, JL and SDH conducted the ice-penetrating radar survey and data processing. IO and KK measured isotopes of gas components. FR, ZTL, WJ, and GMY performed the $^{81}$Kr age dating. SK, JM, and YH measured the stable water isotopes.

**Competing interests**

The authors declare that they have no conflict of interest.

**Acknowledgments**

This study is based on Giyoon Lee's (first author) 2024 doctoral dissertation at Seoul National University. We thank Sang-Young Han, Yoojung Yang, Youngjoon Jang, and Yeongjun Ryu for ice collection in the Elephant Moraine and Reckling Moraine region. We also thank Nayeon Ko, Jinhwa Shin, Junghwa Hwang, and Kwangjin Yim for their laboratory assistance and technical support. We acknowledge the Norwegian Polar Institute's Quantarctica package. Lastly, we would like to thank Editage (www.editage.co.kr) for English language editing.

**Financial support**

This study received financial support from the National Research Foundation of Korea (NRF) (grant no. RS-2024-00449415; RS-2023-00278926; RS-2023-00291696). The Ministry of Science and Technology of China (MOST), Innovation Program for Quantum Science and Technology (2021ZD0303101), the National Natural Science Foundation of China (T2325024, 41727901). This work was also supported by Korea Polar Research Institute (KOPRI) grant funded by the Ministry of Oceans and Fisheries (KOPRI PE25100).

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
