# Peer review of "A 320,000-year-old blue ice identified at the surface of the Elephant Moraine region, East Antarctica"

_EGUsphere, 2025_

## Author Response (AR1)

We sincerely thank the reviewers, Yuzhen Yan and Michael Dyonisius, for their very helpful comments and for thoroughly reviewing the manuscript. The comments were very valuable and helpful in improving the clarity and quality of the manuscript. We have included all the comments and responded to them in detail below. Line numbers refer to the tracked-changes version of the revised manuscript.

**Comments from Yuzhen Yan (Reviewer #1),**

Lee and co-authors present the results of geochemical analyses performed on surface/shallow blue ice samples from Elephant Moraine region, East Antarctica. These data are then used to evaluate the potentials of this region as a paleoclimate archive that extends into the mid-Pleistocene Transition (MPT). Overall, I find the manuscript interesting and worth publications after minor revisions. The merits are two-folded. First, the age (350 ka) of the shallow samples represents an exciting development. Second, the peculiar observations of the altered gas composition near the surface poses a series of questions that warrant further investigation. To better interpret the blue ice records beyond the 800 ka, it is imperative that the glaciological nature of blue ice samples be fully understood.

*Disclosure before detailed comments: I am collaborating with a few of the authors (Zheng-Tian Lu, Wei Jiang, and Guo-Min Yang) on another blue ice project, so I will refrain from evaluating the part pertaining to krypton-81 dating. Rather, my review will focus on chemical measurements such as stable water isotopes and greenhouse gases.

Line 42: "Shallow ice core drilling in BIAs has also been conducted as part of this initiative" It would be nice to give some examples.

➢ Line 45–46: We changed the sentence to "Shallow ice core drilling in Allan Hills BIA has also been conducted as part of this initiative (Yan et al., 2019; Higgins et al., 2025)."

Line 43-44: "The total area of BIAs in Antarctica is estimated to be 234,549 km2, accounting for approximately 1.67 % of the Antarctic continent (Hui et al., 2014)." This sentence seems disconnected from the text before and after it. It would be better to move this sentence to the first paragraph, after this sentence in Line 34: "… outcrops at the surface of the ice sheet in so-called blue-ice areas"

➢ Line 35–36: We moved the sentence as suggested.

Line 52-56: The goal stated here points to the retrieval of ice core in continuous stratigraphy, but given the scope

of the current manuscript it is hard to evaluate if the blue ice record at EM and RM is continuous after all. Finding another blue ice field with >1 Ma samples is a nice complement to the studies at Allan Hills. Furthermore, regardless of the stratigraphic continuity, getting the chronology is a must for any blue-ice studies, so I suggest remove the goal that you can't reasonably accomplish with the current work.

➤ Line 61–62: We deleted the sentence "To better understand the causes of the MPT, it is essential to obtain an ice core continuously spanning this period with pristine gas records.".

➤ Line 63: We revised "…Antarctic BIAs may provide promising sites for…" to "…Antarctica may provide additional promising sites for…".

Line 135-136: The wording here sounds as if the FID can detect CO2 and the conversion to CH4 is for the sake of sensitivity. This is not correct, as FID is highly selective to hydrocarbons (methane included). CO2, by nature, cannot be measured by FID.

➤ Line 152–153: We changed the sentence to "For this, we used a Ni-catalyst to convert $CO_2$ into $CH_4$ before reaching the detector (Ahn et al., 2009; Shin et al., 2022).".

Line 140: The amount of ice seems larger than what is normally required to achieve GC-FID measurements. Is this due to the lower gas content?

➤ In our knowledge, approximately 40-50 g of ice is typically required for $CH_4$ measurement in discrete samples. The 45-56 g range used in this study is slightly larger but we think it is within the normal range of ice required.

Line 164: "potentially contributing to differences in the provenance of blue ice between the EM and RM regions." So far there is no evidence of the provenance of blue ice in the EM and RM regions being different. I suggest move this explanation to the section where you could confidently draw the conclusion (e.g. 3.3 and 3.5).

➤ Line 183–184: Instead of moving the sentence, we revised the sentence to "This topographic feature may act as a barrier for ice flowing from the EM to RM region".

Figure 3: I recommend re-draw this figure and divide it to three panels arranged according to cores rather than the properties being measured. The reason is that it would be useful to investigate any co-variations between CO2 and CH4, which cannot be achieved using the current version.

➢ We re-drew Figure 3 as suggested.

[Figure]

➢ Line 234–235: We deleted "The gray shaded area indicates natural concentration range of atmospheric $CO_2$ and $CH_4$ during the past 800 kyr.".

➢ Line 235–236: We added "Brown lines indicate the dust band identified in the EM-B and EM-C ice cores. The dust band in EM-B has a dip of 59°, whereas the dust band in EM-C is parallel to the surface (Jang et al., 2017).".

Line 231: The d-excess could also be used to indicate the different provenance.

➢ Line 262–263: We added "The significantly different d-excess values in Elephant Moraine Main Icefield may indicate that the provenance of the blue ice differs from that of the other icefields.".

Line 244-245: The question is how melting could happen a few meters below the surface. Although the maximum

austral summer air temperature is −9.5 °C, direct heating during the austral summer could lead to partial melting at the surface of BIAs.

➢ Line 275: We toned down the sentence by changing "suggests" to "could indicate".

➢ Line 278: We added "direct heating by". The revised sentence is as follows:

Despite the absence of visible melt layers, line scanning is warranted to detect any potential small-scale melt layers that could result from direct heating by sunlight penetration into blue ice (Cooper et al., 2021).

Line 248-250: The possibility of gas loss during storage and exposure would lead to more depleted Ar/N2 and O2/N2 ratios, contrary to your observations.

➢ Line 282–283: We added "However, gas loss in bubbly ice samples lead to more depleted $\delta O_2/N_2$ and $\delta Ar/N_2$ values, which is contrary to our observations discussed in Sect. 4.1.2.".

Section 4.1.3: Because greenhouse gas concentrations are one of the most interesting properties in polar ice cores, it is necessary to dive deeper to the issue of altered greenhouse gases. Here, modern air intrusion and melting are discussed, but other ways of producing and consuming GHGs warrant discussion. For instance, could the high CO2 be due to the in situ production? The low concentration of CH4 is somewhat more puzzling and harder to explain.

➢ Line 310–317: We added "Altered GHG concentrations in blue ice are also identified in other BIAs in Antarctica and several hypotheses have been suggested to explain the alteration (Turney et al., 2013; Baggenstos et al., 2017; Dyonisius et al., 2023). For example, based on carbon isotopic ratio measurement of $CO_2$, elevated $CO_2$ concentrations have been attributed either to in-situ production from organic compounds or to ice contamination during sampling, transport, and storage (Turney et al., 2013). Another study, which measured carbon isotopic ratio of $CH_4$ has proposed microbial methanotrophic activity as a potential explanation for the low $CH_4$ concentrations observed in blue ice (Dyonisius et al., 2023). Furthermore, elevated GHG concentrations in blue ice could be attributed to microbial activity (Stibal et al., 2012; Baggenstos et al., 2017). However, more rigorous investigation, including analyses of stable isotopes of GHGs, is required to understand its alteration mechanisms in blue ice.".

Line 268: If you revise Figure 3 and draw it according to cores, then the presence of the dust band can be marked on the new Figure. This way you could further add features of the cores to the figure to explain the altered gas

composition.

➤ We re-drew Figure 3 as suggested.

Line 275: the >300 ppm CO2 in EM-B core (around 9-m depth) also warrant further investigations, especially later in the text only the top 3 meters of the data are believed to be not pristine.

➤ Line 308–309: We added ", and the elevated $CO_2$ concentration at depth of approximately 9 m in EM-B".

**Comments from Michael Dyonisius (Reviewer #2),**

**General comments**

In an initial effort to characterize the BIA (Blue Ice Area) of the Elephant Moraine region, Lee et al. presented gas analyses (mainly, 81Kr, CO2, CH4, N2O, d15N, d18Oatm, O2/N2, and Ar/N2) from 3 shallow (<10m) cores ("EM-B", "EM-C", "EM-K"). Furthermore, they also presented water isotopes data (d18Oice, dD, and d-excess) from near surface ice collected across a transect on nearby the Reckling Moraine region. Finally, these discrete ice core measurements are complemented by an airborne radar survey showing the ice thickness and bedrock elevation of the study site. The main result of the study is the exciting discovery of 350 kyr-old ice at the near surface of Elephant Moraine using 81Kr dating method. Drawing parallels from Allan Hills site (where ~200kyr ice was initially found at the surface, and subsequent deep drilling resulted in >6Myr ice) the authors then argued that it is also possible that the Elephant moraine region might have older ice (mainly spanning the MPT – Mid-Pliocene-Transition) at greater depths. The authors (I presume) initially aimed to use measurements of well-mixed atmospheric gases (mainly, CH4, d18Oatm, but maybe even CO2, and also O2/N2 for tie-in to insolation) to further refine the stratigraphy and age scale, but unfortunately some of the gas compositions in Elephant Moraine seem to have been altered by processes that we (gas people in ice core as a whole, including the authors) do not fully understand.

The finding of another BIA containing old >350kyr is significant, and the study site (Elephant Moraine region) is clearly worth revisiting for future follow-up studies. I also believe that postdepositional alterations in the gas compositions are themselves worthwhile results to document so one day we can understand all the processes in glacial ice that can potentially alter gases in ice cores. As such, I recommend this manuscript for publication, after

some relatively minor revision as noted below.

First, I have some general comments/questions:

1. As currently written on the manuscript, the authors seem to strongly believe that ice spanning the MPT can be retrieved from Elephant Moraine (e.g., line 26 in the abstract: "Given these findings, we _expect_ that ice spanning the MPT period can be retrieved from the Elephant Moraine Main Icefield with only a few hundred meters of drilling."). I think "expect" is a fairly strong word to put on the conclusion of the abstract, especially when the depths "few hundred meters" are also stated.

   The comparison with Allan Hills (e.g., line 287 onwards) is fair, but in my opinion, Allan Hills also had a couple more things going on. Glaciological survey of Allan Hills (Spaulding et al., 2012) provided the legwork showing that ice in Allan Hills BIA flow into a Nunatak and stagnated (there were barely any measurable horizontal velocity at the surface over years of surveys) – so people have long suspected that there is really old ice at Allan Hills. Even under this scenario, when they eventually retrieved the deep core, the ice show almost no change in age for the first ~100m and then a seemingly random age discontinuity (which to my knowledge, was not predicted based on any surface ice/dust stratigraphy expression) into >1Myr ice (Higgins et al., 2015; Yan et al., 2019).

   I personally don't quite see any robust evidence that guarantees MPT ice in Elephant Moraine: (a) the ice flow is relatively unimpeded at 1-5 m/yr, (b) the shallow cores do not show significant change in age vs. depth. So, I would encourage the authors to either provide stronger evidence/reasoning for MPT ice at Elephant Moraine, or tone down the language/expectation slightly into "maybe"s and "could"s.

   ➢ Line 26–28: We revised the sentence as "Given these findings, ice spanning the MPT period could be retrieved from the Elephant Moraine Main Icefield with only a few hundred meters of drilling.".

2. Post-coring gas-loss is a significant factor that could potentially alter O2/N2, Ar/N2, and d18Oatm – as the authors clearly understood and discussed extensively in the manuscript. I think a few additional descriptions regarding sample transport and storage are warranted. Was there any temperature logger in the ice core box to ensure that the samples never see temperatures higher than -30C during transport? What is the temperature of the freezer where the ice samples are stored (presumably at SNU?). How much time passed between the initial retrieval (2016/2017 austral summer) and O2/N2, Ar/N2, d15N, and d18Oatm analyses at NIPR? These are worth describing given the context of the data presented in

this manuscript just to rule out potential handling problems affecting the gas measurements.

➢ About the temperature logger during transport:

Yes, we tracked the temperature with a logger during transport. To the best of our recollection, the temperature remained low enough (around −20 °C to −15 °C) to prevent melting. However, unfortunately, the logger data were no longer available.

➢ Line 81–82: We added "Ice samples were stored at Seoul National University (SNU) and Korea Polar Research Institute (KOPRI) at −20 °C until analysis.".

➢ Line 124: We added "on December 2019".

➢ Line 126–127: We added "Until analysis on December 2022, the ice samples were stored at NIPR at around −30 °C.".

3. At the end, the authors used the low d18Oatm measurement (of -0.105 permil in EM-K core at 4.9m depth) to argue that the average age for this particular core is MIS9 (~320kyr) instead of MIS10 (350 kyr) as suggested by 81Kr dating, and in direct contradiction with the manuscript's own title "A 350,000-year old blue ice identified …". For example, in line 284: the authors mentioned "Similarly, the ice from EM-K core is unlikely to correspond to the period of MIS 10–11 and may instead be from MIS 9 (Fig. 5)." Personally, I'm not sure if I would believe the potentially gas-loss altered d18Oatm data point more than 81Kr date, but if the authors do (as written in the conclusion section), then the title of the manuscript needs to be changed to "~320,000 years old blue ice identified." Also, I'm not sure why the authors are emphasizing specifically "MIS9" in Section 4.2 and Section 5 (it could've also been MIS11 which is on the other end of 81Kr age uncertainty).

➢ Line 1: We changed "350,000-year-old" to "320,000-year-old".

➢ Line 322–332: We revised this part as follows:

For this comparison, we used the measurement results from depths greater than 3 m, excluding those that fall outside the natural range during the past 800 kyr; we also excluded the elevated GHG concentrations observed at depths of approximately 9 m in EM-B and 5 m in EM-K. Assuming that the original values fall within the average and standard deviation of the measurement results used, EM-B and EM-C cores do not correspond strongly to MIS 5e as the $CH_4$ concentration and $\delta^{18}O_{atm}$ values

significantly differ from that period (Fig. 5). Similarly, the ice from the EM-K core is unlikely to correspond to MIS 10–11 and may instead be from early MIS 9, as all three measured gas components ($CO_2$, $CH_4$, and $\delta^{18}O_{atm}$) are consistent with values observed during early MIS 9 (Fig. 5). We consider the measurements from the EM-K core to be more reliable than those from the EM-B and EM-C cores, as TAC values are closer to that of typical deep Antarctic ice cores (Table 3).

➢ Line 356: We added "early".

**Minor/line comments.**

Two minor general comments before the line comments:

1. A couple more reasons to study BIA that the authors can mention in the abstract/introduction is that (a) the old ice is often found at near surface, and thus it is easier to access logistically and (b) easily accessible old ice in BIA provides the testbed for measurements and development of exotic tracers that are not yet possible/too risky to do on traditional deep ice core.

➢ Line 36–38: Point (a) was already described in the introduction.

➢ Line 38–39: We added "In addition, easily accessible old ice in BIAs offers a valuable testbed for developing and applying novel exotic tracers that are currently too risky or impractical to use in conventional deep ice cores." for the point (b).

2. For the alterations in GHG ($CO2$, $CH4$, $N2O$) – it might seem obvious, but still it is worth mentioning that for future studies analyses of their stable isotopes can help shed light on the processes responsible (whether it is microbial production, consumption, or in situ production by organic/inorganic impurities).

➢ Line 310–317: We added "Altered GHG concentrations in blue ice are also identified in other BIAs in Antarctica and several hypotheses have been suggested to explain the alteration (Turney et al., 2013; Baggenstos et al., 2017; Dyonisius et al., 2023). For example, based on carbon isotopic ratio measurement of $CO_2$, elevated $CO_2$ concentrations have been attributed either to in-situ production from organic compounds or to ice contamination during sampling, transport, and storage (Turney et al., 2013). Another study, which measured carbon isotopic ratio of $CH_4$ have proposed microbial methanotrophic activity as potential explanations for the low $CH_4$ concentrations observed in blue ice (Dyonisius et al., 2023). Furthermore, elevated GHG concentrations in blue ice could be attributed to microbial activity

(Stibal et al., 2012; Baggenstos et al., 2017). However, more rigorous investigation, including analyses of stable isotopes of GHGs, is required to understand its alteration mechanisms in blue ice.".

Line 47: "The oldest ice is found in the Allan Hills BIA, where the surface ice age ranges from 90 kyr BP to 250 kyr BP (Before Present) (Spaulding et al., 2013), and ice at depths of 200 m dates back to approximately 6 Myr BP (Higgins et al., 2025)." I would mention Mullins ice first. Actually, based on 40Ar/39Ar tephra dating, people have claimed that ice in Beacon Valley (which is downstream of Mullins Glacier) to be up to 8Myr old (Marchant et al., 2002) – but this number has been questioned (whether the tephra there is pristine deposition or has been reworked/blown from elsewhere in the Dry Valleys). The d40/36Ar dating showed much younger age (Yau et al., 2015) of 1.6 Myr – but Yau et al. discussed that there is plenty evidence for modern air intrusion from visual cracks and low TAC in these ice, so the 1.6 Myr 40/36 age for Mullins Glacier is a conservative lower limit for how old the ice can be. Another problem with Mullins Glacier and Beacon Valley (other than modern contamination from cracks and microbes altering the CO2) that can be mentioned is that Mullins is a rock glacier, there is no "blue ice", so access (e.g., drilling through both rock and ice) is extremely difficult and environmentally destructive (whole polygon has to be excavated). Thus, Allan Hills is the oldest site we have for "clean" easily accessible ice.

➢ Line 54–65: We changed this part as follows:

Very old ice has also been identified in rock glaciers in Antarctica (Table 1). For example, ice at depths of 3–32 m from the Mullins Glacier has been dated to 1.6 Myr BP (Yau et al., 2015) and ice found in Beacon Valley, which is downstream of Mullins Glacier, up to 8.1 Myr BP (Marchant et al., 2002) (Table 1). The estimated gas age of ice at Mullins Glacier is considered a lower bound because the analyzed air likely represents a mixture of ancient and recent atmosphere (Yau et al., 2015). The age constraint for ice in Beacon Valley, based on $^{40}Ar/^{39}Ar$ tephra dating, has been questioned due to the possibility for reworking and re-transportation of the tephra. Based on the discovery of pre-MPT ice in the Allan Hills BIA and Mullins Glacier, Antarctica may provide additional promising sites for recovering such an ice core by shallow drilling. To identify potential sites, chronological studies of the surface ice must first be conducted.

➢ Line 72: We also changed Table 1 as follows:

**Table 1.** Age constraints of Antarctic blue-ice areas (BIAs) and rock glaciers[*].

| Blue-ice areas | Age (kyr BP) | Location | References |
| --- | --- | --- | --- |

| | | | |
|---|---|---|---|
| Meteorite City Icefield | 101, 108 | 76.25° S, 156.56° E | This study |
| Elephant Moraine Main Icefield | 320 | 76.32° S, 157.20° E | This study |
| Allan Hills | 90–250, 2700, 6000 | 76.73° S, 159.36° E | Spaulding et al. (2013), Yan et al. (2019), Higgins et al. (2025) |
| Frontier Mountain | <50 | 72.98° S, 160.33° E | Folco et al. (2006) |
| Grove Mountains | 143 | 72.99° S, 75.22° E | Hu et al. (2024) |
| Larsen Glacier | 6–25 | 74.93° S, 161.60° E | Lee et al. (2022) |
| Mt. Moulton | 105–136, 496 | 76.67° S, 134.70° W | Dunbar et al. (2008), Korotkikh et al. (2011) |
| Mullins Glacier* | 1600 | 77.88° S, 160.54° E | Yau et al. (2015) |
| Beacon Valley* | 8100 (?) | 77.85° S, 160.59° E | Marchant et al. (2002) |
| Nansen | <130 | 72.75° S, 24.50° E | Zekollari et al. (2019) |
| Patriot Hills | 1–80, 130–134 | 80.30° S, 81.35° W | Turney et al. (2020) |
| Scharffenbergbotnen | <11 | 74.56° S, 11.05° W | Sinisalo et al. (2007) |
| South Yamato | 55–61 | 72.08° S, 35.18° E | Moore et al. (2006) |
| Taylor Glacier | 9–133 | 77.75° S, 161.80° E | Buizert et al. (2014) |

Line 57: "In this study, we investigated icefields in the Elephant Moraine (EM) and Reckling Moraine (RM) regions, focusing primarily on constraining the age of blue ice in the EM region." Refer to Figure 1. Also, since this is the first time the study sites are mentioned, give the coordinates.

➢ Line 66–67: We added the coordinates. The revised sentence is as follows:

In this study, we investigated icefields in the Elephant Moraine (EM) (76.32° S, 157.20° E) and Reckling Moraine (RM) (76.24° S, 158.39° E) regions, focusing primarily on constraining the age of blue ice in the EM region (Fig. 1).

Line 66: "During the 2016/17 austral summer, shallow ice cores (5–10 m in length) were collected from the icefields within the EM region (Fig. 1) (Jang et al., 2017)." I would introduce here specifically 3 shallow cores "EM-B, EM-C, and EM-K" are retrieved. Also mention the drill/core diameter and mention that gas analyses ($81Kr$, $CO_2$, $CH_4$, $N_2O$ mole fraction, $d15N$, $d18Oatm$, $O_2/N_2$, and $Ar/N_2$) are conducted on the shallow cores (refer to respective methods section).

➢ Line 75–79: We revised this part as follows:

During the 2016/17 austral summer, shallow ice cores (5–10 m in length and 10 cm in diameter) were collected from the icefields within the EM region (Fig. 1) (Jang et al., 2017). In this study, three shallow cores (EM-B, EM-C, and EM-K) were used for gas analyses ($^{81}$Kr, $^{85}$Kr, $\delta^{15}$N-N$_2$, $\delta^{18}$O-O$_2$, $\delta$O$_2$/N$_2$, $\delta$Ar/N$_2$, CO$_2$, CH$_4$, and N$_2$O). Refer to Sect. 2.3 for Kr measurements, Sect. 2.4 for isotopic ratio measurements of major gas components, and Sect. 2.5 for greenhouse gas concentration measurements, respectively.

Line 66: "Additionally, during the 2018/19 austral summer, 70 surface ice samples (5–10 cm depth) were collected along a 700 m transect at 10 m intervals from the icefield in the RM region (Fig. 1)." Mention that the surface transect samples are only measured for water isotopes and refer to Section 2.6

➢ Line 79–81: We revised this part as follows:

Additionally, during the 2018/19 austral summer, 70 surface ice samples (5–10 cm depth) were collected along a 700 m transect at 10 m intervals from the icefield in the RM region and were analyzed for stable water isotopes (Fig. 1) (Sect. 2.6).

Line 77: Figure 1. Add the arrow showing flow direction in inset (c) – to show what the surface transect is somewhat perpendicular to flow direction. Also, if this was intentional (the transect being perpendicular to ice flow to maximize potential changes in age vs. distance) mention it.

➢ Ice flow vectors were not shown for the Reckling Moraine Icefield because relevant ice flow vectors were not available. Also, the sampling transect being somewhat perpendicular to the ice flow direction was not intentional.

Line 109: "Six ice samples were cut from the EM ice cores and sent to the National Institute of Polar Research (NIPR) in Japan for the simultaneous measurement of O2, N2, and Ar isotopes using a dual-inlet mass spectrometer (Thermo Fisher Delta V)." This applies to other experimental method descriptions (Section 2.3, 2.4, 2.5, 2.6), for conciseness I would say "samples were measured for x tracers, at y institution following Z. et al." and remove the sentence at the end (e.g., line 114-115: "Further details on the procedure can be found in the study by Oyabu et al. (2020)."

➢ Line 117–119: We changed the method description as follows:

$^{81}$Kr analysis was performed using the Atom Trap Trace Analysis (ATTA) method, and $^{85}$Kr was also measured to quantify the potential contamination from modern air, following Tian et al. (2019) and Jiang et

al. (2012).

Based on ice core availability, six ice samples were cut from the EM ice cores and sent to the National Institute of Polar Research (NIPR) in Japan on December 2019 for the simultaneous measurement of $O_2$, $N_2$ isotopes, and $O_2$, $N_2$, Ar molecular ratios using a dual-inlet mass spectrometer (Thermo Fisher Delta V) following Oyabu et al. (2020).

The $CO_2$ concentrations in the EM blue ice (EM-B, EM-C, and EM-K) were measured at SNU following Shin (2014) and Lee et al. (2022).

The concentrations of $CO_2$, $CH_4$, and $N_2O$ in several ice core samples were also measured along with gas isotopes ($\delta^{15}N$-$N_2$, $\delta^{18}O$-$O_2$, $\delta O_2/N_2$, and $\delta Ar/N_2$) at the NIPR using wet extraction method following Oyabu et al. (2020).

Stable water isotopes ($\delta^{18}O_{ice}$ and $\delta^2H_{ice}$) of the surface RM blue ice (approximately 5–10 cm depth) and EM-K core were measured at the Korea Polar Research Institute (KOPRI) using cavity ring-down spectroscopy (CRDS, Picarro L2130-i) (Fig. 1) following Lee et al. (2022).

Also, for the convenience of the readers, for the methods described please provide the standard measurement metrics (e.g., repeatability for d15N2, d18Oatm, O2/N2, Ar/N2, and the GHG mole fractions).

➢ Line 125–126: We added "The reproducibility for $\delta^{15}N$-$N_2$, $\delta^{18}O$-$O_2$, $\delta O_2/N_2$, and $\delta Ar/N_2$ are 0.006 ‰, 0.011 ‰, 0.09 ‰, and 0.12 ‰, respectively (Oyabu et al., 2020).".

➢ Line 154–156: We added "The uncertainty of the $CO_2$ concentration measurement is defined as the standard deviation of the $CO_2$ measurement results from the control group (average of intra-day standard deviation of the control group: $0.6 \pm 0.6$ ppm).".

the control group: 3.3 ± 1.4 ppb).".

Also, specifically for this sentence (line 109), "Ar isotopes" is a bit misleading because it implies that m/z 40, 38, 36 are measured for d40/36 Ar dating. I would instead say "simultaneous measurements of O2, N2 isotopes, and O2, N2, Ar molecular ratios …"

➢ Line 123–125: We changed the method description as follows:

Based on ice core availability, six ice samples were cut from the EM ice cores and sent to the National Institute of Polar Research (NIPR) in Japan on December 2019 for the simultaneous measurement of $O_2$, $N_2$ isotopes, and $O_2$, $N_2$, Ar molecular ratios using a dual-inlet mass spectrometer (Thermo Fisher Delta V) following Oyabu et al. (2020).

Line 106: Table 2. Currently, the depth ranges used for 81Kr measurements (as well as noble gas measurements conducted at NIPR shown in Table 3) seem quite random. Please describe the method and thinking behind the gaps (is it to avoid unusually high CO2 and CH4 mole fraction at ~5m depth? Is it to avoid visible cracks? Is it to bracket the mid-depths? Or is it just whatever remaining ice from shallow cores which is available).

➢ Line 112–113: We already mentioned why the depth ranges are quite random "Because of ice core availability, we mixed different depth ranges for EM-B and EM-K (Table 2).".

➢ Line 123: We added "Based on ice core availability,".

Line 126: "However, due to insufficient measurements for gas loss correction, we could not apply gas loss correction in this study (Landais et al., 2003; Capron et al., 2010)." For gas-loss correction method that is specifically applied to BIA samples please also cite Baggenstos et al. (2017) (Eq.2 in the paper).

➢ Line 143: We added the suggested reference.

Line 166: Figure 2 caption. Mention that the bedrock elevation shown here is AMSL – above mean sea level.

➢ Line 187: We added "(AMSL)".

Line 188: "The gravity-corrected …" This is a minor thing, but I don't think "gravity-corrected" is quite right. Just be descriptive and say "The dO2/N2 and dAr/N2 corrected for gravitational fractionation […] "

➢ Line 210: We changed it as suggested.

Line 193: "3.4 Greenhouse gases (CO2, CH4)." I know most people don't care about N2O, but N2O is also a greenhouse gas and happen to be measured in this study, just briefly mention that similar to CO2 and CH4, N2O mole fraction measured are in line with warm interglacial values, but N2O in BIA might be affected by in-situ production from dust (Schilt et al., 2014).

➢ Line 229–230: We added "Several measured $N_2O$ concentrations (Table 3) were in line with warm interglacial values, but they might have been affected by in-situ production from dust (Schilt et al., 2014)." as suggested.

"Figure S1. Measured stable water isotope values and d-excess of Reckling Moraine (RM) blue ice." Mention these are from surface/transect samples.

➢ We changed the caption to "Figure S1. Measured stable water isotope values and d-excess of surface ice collected from the transect at Reckling Moraine (RM) Icefield.".

Line 217: "Since the typical glacial-interglacial δ18Oice difference in East Antarctica is 5–6 ‰ (Stenni et al., 2010), the very wide δ18Oice range observed in RM blue ice (from −51.2 ‰ to −34.9 ‰) suggested significant differences in provenance of blue ice." I'm not a water isotope expert, so I might be wrong, but seems like the glacial-interglacial d18Oice range of 5-6 permil cited here only applies to ice cores from way up in the ice sheet (dome site and traditional deep ice core). This is likely not the case for the accumulation site of Elephant Moraine. Being (presumably) located on the flank and even near the coast of East Antarctic ice sheet, the accumulation site of Elephant Moraine would/might encounter large change in elevation (and thus water isotope values) as well as temperature, especially over periods of kyrs. Can this be an alternative explanation instead of difference in provenance?

➢ If the wide range is not due to difference in blue ice provenance, it may reflect of a large change in surface temperature and/or elevation. As the cause of the observed wide range in water stable isotope is uncertain, we added a discussion noting that both surface elevation and/or temperature changes should be considered.

➢ Line 243–248: We changed this part as follows:

Since the typical glacial-interglacial $\delta^{18}O_{ice}$ difference in East Antarctica is 5–6 ‰, based on conventional deep ice cores (Stenni et al., 2010), the very wide $\delta^{18}O_{ice}$ range observed in RM blue ice (from −51.2 ‰ to −34.9 ‰) may suggest significant differences in provenance of blue ice. Alternatively, since the accumulation site of RM region is likely located on the flank of the East Antarctic ice sheet, it may have experienced large changes in surface elevation and/or temperature. Hence, if the provenance is not significantly different, such

a wide range could indicate large changes in surface elevation and/or temperature.

Line 248: "Further studies should also investigate the possibility for gas loss during storage, as the ice cores were kept at temperatures above −50 °C (Oyabu et al., 2021)." This sentence is a bit confusing. I think "the ice cores" here in the sentence refer to ice core samples of this study? I would paraphrase to "Studies have shown that gas loss during storage occurs when ice core samples were kept at temperatures above –50 °C (Oyabu et al., 2021). The ice core samples presented in this manuscript were kept at x degrees for y years."

➢ Line 278–281: We revised it to "Studies have shown that gas loss during storage occurs when ice core samples were kept at temperatures above –50 °C (Oyabu et al., 2021). After ice core drilling, samples presented in this study were kept at −20 °C or −30 °C for several years until analysis.".

Line 267: "A study using blue ice in Pakitsoq, western Greenland also revealed that enriched CH4 values were correlated with visible dust bands, but the mechanism remained unclear (Petrenko et al., 2006)." Recent studies (e.g., Lee et al., 2020; Mühl et al., 2023) have investigated the in-situ production of CH4 associated with high dust content.

➢ In our understanding, the two papers you suggested (Lee et al., 2020; Mühl et al., 2023) are about $CH_4$ production due to different gas extraction method, not in-situ $CH_4$ production. Also, the ice samples used are not blue ice samples. Therefore, we think the two papers suggested are not available to explain the high $CH_4$ concentrations in near-surface blue ice.

Line 274: "Further investigation required to better understand the cause of GHG alterations in EM blue ice, particularly the very high CO2 concentration (628 ppm) and unusually low CH4 concentration (207 ppb) in the EM-K core." This is not too unusual for BIA. We also often observe CH4 depletion and elevation (often simultaneously) at near surface shallow depths in Taylor Glacier (e.g., Baggenstos, 2015 - Figure 2.3) (Dyonisius et al., 2023 - Figure S11,S12). In Dyonisius et al. (2023) we have d13C-CH4 measurement from surface samples showing that the low CH4 is associated with high d13C-CH4 (which indicate consumption from microbes, which preferentially takes up 12C and leaves the CH4 in the bubbles enriched with d13C). At the old (>50kyr) surface site in Taylor Glacier, at depth ~30m we also see age discontinuity where MIS4 (which is a dusty period) is missing entirely, and replaced by ~5m ice of unknown provenance/age with depleted TAC and depleted CH4 mole fraction (Dyonisius et al., 2023 - Figure S11). Generally, the culprit for alteration in CO2, CH4, and N2O mole fraction in BIA will be melt, microbes, and impurities/dust, often in combination of each other. A good paper to cite regarding

how microbes can potentially alter CO2 mole fraction is Stibal et al. (2012).

➢ Line 310–317: We added more discussion about the GHG alteration processes as follows:

Altered GHG concentrations in blue ice are also identified in other BIAs in Antarctica and several hypotheses have been suggested to explain the alteration (Turney et al., 2013; Baggenstos et al., 2017; Dyonisius et al., 2023). For example, based on carbon isotopic ratio measurement of $CO_2$, elevated $CO_2$ concentrations have been attributed either to in-situ production from organic compounds or to ice contamination during sampling, transport, and storage (Turney et al., 2013). Another study, which measured carbon isotopic ratio of $CH_4$ have proposed microbial methanotrophic activity as potential explanations for the low $CH_4$ concentrations observed in blue ice (Dyonisius et al., 2023). Furthermore, elevated GHG concentrations in blue ice could be attributed to microbial activity (Stibal et al., 2012; Baggenstos et al., 2017). However, more rigorous investigation, including analyses of stable isotopes of GHGs, is required to understand its alteration mechanisms in blue ice.

Line 291: Figure 1. I noticed that for d18Oatm (Figure 1C) for "EM-K" core only the lower values of -0.105 permil is plotted. The shallow EM-K core has d18Oatm of 0.286 permil, so the average for EM-K should be ~+0.1 permil; seems like the higher d18Oatm value is dropped/not plotted. Maybe the reasoning is mentioned somewhere and I missed it.

➢ Line 322–323: We mentioned that for comparison we only used measurements from depths greater than 3 m.

Line 292: "Error bars for CO2, CH4, and δ18Oatm represent 1σ standard deviation." Standard deviation of what, please elaborate. Is it stdev from all samples in the whole length of the shallow core, or standard deviation of repeated measurements (e.g., measurement precision).

➢ Line 340: We added "of the measurement result used".

**References**

Baggenstos, D.: Taylor Glacier as an archive of ancient ice for large-volume samples: Chronology, gases, dust, and climate, 2015. PhD thesis. ProQuest.

Baggenstos, D., Bauska, T. K., Severinghaus, J. P., Lee, J. E., Schaefer, H., Buizert, C., Brook, E. J., Shackleton, S., and Petrenko, V. V.: Atmospheric gas records from Taylor Glacier, Antarctica, reveal ancient ice with ages spanning the entire last glacial cycle, Climate of the Past, 13, 943, 2017.

Dyonisius, M. N., Petrenko, V. V., Smith, A. M., Hmiel, B., Neff, P. D., Yang, B., Hua, Q., Schmitt, J., Shackleton, S. A., Buizert, C., Place, P. F., Menking, J. A., Beaudette, R., Harth, C., Kalk, M., Roop, H. A., Bereiter, B., Armanetti, C., Vimont, I., Englund Michel, S., Brook, E. J., Severinghaus, J. P., Weiss, R. F., and McConnell, J. R.: Using ice core measurements from Taylor Glacier, Antarctica, to calibrate in situ cosmogenic 14C production rates by muons, The Cryosphere, 17, 843–863, https://doi.org/10.5194/tc-17-843-2023, 2023.

Higgins, J. A., Kurbatov, A. V., Spaulding, N. E., Brook, E., Introne, D. S., Chimiak, L. M., Yan, Y., Mayewski, P. A., and Bender, M. L.: Atmospheric composition 1 million years ago from blue ice in the Allan Hills, Antarctica, Proceedings of the National Academy of Sciences, 112, 68876891, https://doi.org/10.1073/pnas.1420232112, 2015.

Lee, J. E., Edwards, J. S., Schmitt, J., Fischer, H., Bock, M., and Brook, E. J.: Excess methane in Greenland ice cores associated with high dust concentrations, Geochimica et Cosmochimica Acta, 270, 409–430, 2020.

Marchant, D. R., Lewis, A. R., Phillips, W. M., Moore, E. J., Souchez, R. A., Denton, G. H., Sugden, D. E., Potter, N., Jr., and Landis, G. P.: Formation of patterned ground and sublimation till over Miocene glacier ice in Beacon Valley, southern Victoria Land, Antarctica, GSA Bulletin, 114, 718–730, https://doi.org/10.1130/0016-7606(2002)114<0718:FOPGAS>2.0.CO;2, 2002.

Mühl, M., Schmitt, J., Seth, B., Lee, J. E., Edwards, J. S., Brook, E. J., Blunier, T., and Fischer, H.: Methane, ethane, and propane production in Greenland ice core samples and a first isotopic characterization of excess methane, Climate of the Past, 19, 999–1025, https://doi.org/10.5194/cp-19-999-2023, 2023.

Schilt, A., Brook, E. J., Bauska, T. K., Baggenstos, D., Fischer, H., Joos, F., Petrenko, V. V., Schaefer, H., Schmitt, J., Severinghaus, J. P., Spahni, R., and Stocker, T. F.: Isotopic constraints on marine and terrestrial N2O emissions during the last deglaciation, Nature, 516, 234–237, https://doi.org/10.1038/nature13971, 2014.

Spaulding, N. E., Spikes, V. B., Hamilton, G. S., Mayewski, P. A., Dunbar, N. W., Harvey, R. P., Schutt, J., and Kurbatov, A. V.: Ice motion and mass balance at the Allan Hills blue-ice area, Antarctica, with implications for paleoclimate reconstructions, Journal of Glaciology, 58, 399406, https://doi.org/10.3189/2012JoG11J176, 2012.

Stibal, M., Šabacká, M., and Žárský, J.: Biological processes on glacier and ice sheet surfaces, Nature Geosci, 5, 771–774, https://doi.org/10.1038/ngeo1611, 2012.

Yan, Y., Bender, M. L., Brook, E. J., Clifford, H. M., Kemeny, P. C., Kurbatov, A. V., Mackay, S., Mayewski, P.

A., Ng, J., Severinghaus, J. P., and Higgins, J. A.: Two-million-year-old snapshots of atmospheric gases from Antarctic ice, Nature, 574, 663–666, https://doi.org/10.1038/s41586019-1692-3, 2019.

Yau, A. M., Bender, M. L., Marchant, D. R., and Mackay, S. L.: Geochemical analyses of air from an ancient debris-covered glacier, Antarctica, Quaternary Geochronology, 28, 29–39, https://doi.org/10.1016/j.quageo.2015.03.008, 2015.